# Tree Species Identification in Urban Environments Using TensorFlow Lite and a Transfer Learning Approach

Diego Pacheco-Prado [1,2,*], Esteban Bravo-López [2,3] and Luis Ángel Ruiz [1]

1   Geo-Environmental Cartography and Remote Sensing Group (CGAT), Universitat Politècnica de València, 46022 Valencia, Spain

2   Instituto de Estudios de Régimen Seccional del Ecuador (IERSE), Vicerrectorado de Investigaciones, Universidad del Azuay, Cuenca 010204, Ecuador; pbravo@uazuay.edu.ec

3   Centre for Advanced Studies in Earth Sciences, Energy and Environment, Department of Cartographic, Geodetic and Photogrammetric Engineering, University of Jaen, 23071 Jaen, Spain

*   Correspondence: diepacpr@upv.es

**Abstract:** Building and updating tree inventories is a challenging task for city administrators, requiring significant costs and the expertise of tree identification specialists. In Ecuador, only the Trees Inventory of Cuenca (TIC) contains this information, geolocated and integrated with the taxonomy, origin, leaf, and crown structure, phenological problems, and tree images taken with smartphones of each tree. From this dataset, we selected the fourteen classes with the most information and used the images to train a model, using a Transfer Learning approach, that could be deployed on mobile devices. Our results showed that the model based on ResNet V2 101 performed best, achieving an accuracy of 0.83 and kappa of 0.81 using the TensorFlow Lite interpreter, performing better results using the original model, with an accuracy and kappa of 0.912 and 0.905, respectively. The classes with the best performance were Ramo de novia, Sauce, and Cepillo blanco, which had the highest values of Precision, Recall, and F1-Score. The classes Eucalipto, Capuli, and Urapan were the most difficult to classify. Our study provides a model that can be deployed on Android smartphones, being the beginning of future implementations.

**Keywords:** Convolutional Neural Network; CNN; ResNet V2 101; smartphones; TensorFlow; Transfer Learning; Tree Inventory





## 1. Introduction

Urban trees benefit citizens, improving their quality of life [1–3]; therefore, their conservation is essential for city administrators [1]. Urban Tree Inventories (UTIs) provide information about species and location and photography of each tree [2,3], which allow sustainable forest management and monitoring of urban trees and gardens [4]. Nevertheless, the costs of carrying out these inventories are usually high, impeding their creation [5] or updating [6], and requiring specialists to identify tree species [7]. Participatory projects such as Tree Inventory on Portland's streets [8] or New York City [9], or private projects such as Cuenca-Ecuador [10] with smartphones [3,10,11] have allowed the collection of these datasets.

In recent years, identifying plant species [12–16], flowers [17], medicinal plants [18], and trees [19] has been achieved using photographs. Examples of this are some mobile applications available in the android market, such as LeafSnap, PictureThis, PlantNet, Blossom, Seek, and others [7]. Plant identification approaches are usually based on analyzing a single leaf followed by a flower [7]. For these image analysis systems that apply Artificial Vision and Deep Learning methods, it is possible to customize models of local flora [20], to assess the health status of trees and plants [16,21] and aid in leaf recognition and classification [22,23] using artificial neural networks.

Several studies have analyzed the application of Deep Learning for forest analysis from different approaches. In [24], Deep Learning algorithms were analyzed for forest resource inventory and tree species identification, implementing neural networks of different architectures. In [25], the use of Machine Learning (ML) algorithms (Random Forest, Support Vector Machines, and ANN Multilayer Perceptron) was analyzed for tree species classification in urban environments through the use and processing of LiDAR data. Although the focus of these studies is different from the one proposed in the present research, it is important to show that the use of quantitative methods based on ML algorithms contributes to the optimization of management, planning, and maintenance of tree inventories in urban areas with complex structures. On the other hand, there are also studies in which Deep Learning was applied explicitly for forest species identification. In [26], plant species mostly coming from western Europe and North America were identified based on images from the online collaborative Encyclopedia of Life https://eol.org/ (accessed on 20 February 2023), using Deep Convolutional Neural Network (DCNN) architectures, such as GoogleNet, ResNet, and ResNeXT. In [19], tree species were also identified using Convolutional Neural Networks (CNNs), but based on images of the bark. The images used were obtained under different conditions and cameras to ensure a diversification of the data. The accuracy obtained was higher than 93%, demonstrating the effectiveness of the application of this methodology. In [27], they used Deep Learning, implementing the UNET architecture (a variant of CNN), to classify tree species using forest images, considering additional important attributes for the stands and parcels from which the images were obtained, such as elevation, aspect, slope, and canopy density, and obtained accuracy results above 80% based on different validation metrics such as accuracy, Precision, and Recall. In [28], a more specific analysis was performed, detecting and identifying plant diseases in real-time using a CNN-based architecture on an integrated platform by analyzing the leaves. The accuracy of the results obtained was higher than 96%. All these investigations demonstrate that methodologies focused on Machine Learning and Deep Learning, such as CNNs, can be used to identify forest species from different perspectives and datasets with high accuracy.

CNN is nowadays the most used Neural Network for image classification. It consists of a stack of layers encompassing an input image, performing a mathematical operation, and predicting the class or label probabilities in the output [29]. This can be implemented in the TensorFlow framework [30], the most stable and complete library for the Python language [31].

Image classification models using CNNs can have millions of parameters, which means that training them from the beginning requires a large amount of labeled input data [19,32,33] and high computing power [34]. Another problem is that a limited number of samples can generate overfitting problems in CNN models, which consist of the poor generalization capacity of the models to new information [32]. This can be reduced through two techniques: Transfer Learning (TL) and Data Augmentation (DA). TL is a process that allows the creation of new learning models by adjusting once-trained neural networks [34,35]. DA generates synthetic samples from existing data using transformations, such as image rotation, color degradation, and others, although it can cause overfitting in minority classes [36]. TensorFlow Hub offers a pre-trained and ready-to-use model for TL [37] that simplifies training models using a custom dataset [38]. Additionally, the TensorFlow Lite Model Maker function allows for these models to be deployed for mobile devices.

The trained models can run inference on the client's devices when using TFLite files for our Java-developed Android application. AgroAId is an example; it was built to identify combinations from a plant leaf image input, recognizing 39 species and disease classes on devices with limited computing and memory resources [39]. Finally, the relevant pixels for the decision to classify the image by CNN can be visualized through the GRAD-CAM (Gradient-weighted Class Activation Mapping) method, which highlights the areas of the image in the form of heat maps [32,40,41].

If we consider that most of the mobile applications available require an Internet connection to perform the identification of plants or forest species [7], that means that the inference is performed in a medium external to the mobile device and it limits its use in

disconnected environments; thus our main contribution is to evaluate a methodology that trains a model that can be used offline and is based on Transfer Learning methodology due to the limited number of samples available compared to other free datasets. On the other hand, currently available mobile applications for plant identification focus on leaves or fruits because these elements are the most easily recognizable and distinctive parts of a plant. In the case of trees, the reviewed works evaluate photographs of the trunk or images with environmental noise. Nevertheless, our work analyzed trees based on their general appearance in order to identify those whose characteristics can be easily evaluated by a CNN, thus allowing for their identification.

This research applies and evaluates a Transfer Learning for automated identification of the most common tree species in the urban area of Cuenca (Ecuador), using computer vision techniques based on smartphone images of the local flora from the Tree Inventory of Cuenca (TIC), and can be deployed using Android smartphones.

## 2. Materials and Methods

### 2.1. Study Area

Cuenca is located in Ecuador's central/southern area (Figure 1) at a mean elevation of 2500 m above sea level (m.a.s.l.). According to information from the Municipal Cleaning Company of Cuenca (EMAC-EP), in 2017 there were around 208 parks (public green areas) within the urban perimeter of the city [4], with the largest and the most important green areas in El Paraíso, Miraflores, and La Madre.

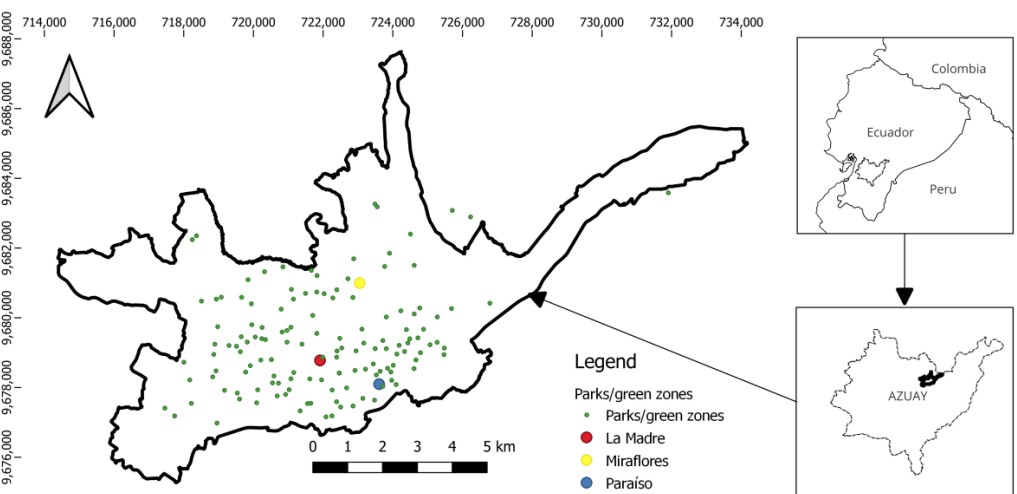

**Figure 1.** Location of the city of Cuenca and its parks.

In Cuenca (Ecuador), the University of Azuay, through the Institute for Sectional Regime Studies of Ecuador—IERSE https://ierse.uazuay.edu.ec (accessed on 5 February 2023), —and the Azuay Herbarium https://herbario.uazuay.edu.ec/ (accessed on 5 February 2023), built the Tree Inventory of Cuenca and published the results on the official website https://gis.uazuay.edu.ec/iforestal/ (accessed on 1 March 2023), [10]. Until June 2022, it contained 12,688 trees with information about taxonomy, origin, crown and trunk structure, phenology, forest management, potential problems, and two photographs of each tree. Of these trees, 52.1% corresponded to introduced species, 47.7% are native, and less than 1% are endemic species. The most frequent species are Fresno o Cholan, Sauce, Urapan, Jaracaranda, Capuli, and Cepillo Blanco.

### 2.2. Methodology

The methodology was implemented in four phases (Figure 2): (1) assessment of existing classes in the TIC and selection of those with the largest number of images, with trees over two meters height. (2) Image selection by activation maps and balancing of the number of images per class using an image rotation operation to reach the minimum

image number. (3) Training the models with the selected images, evaluation, and producing classification reports. (4) Deployment of the top-ranked TFLite model on Android devices.

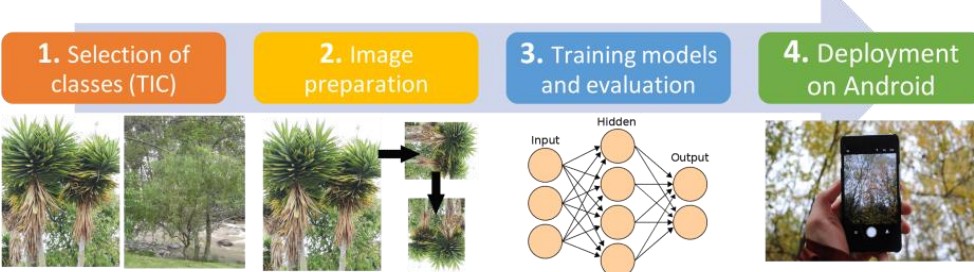

**Figure 2.** Overlay workflow of the methodology followed to create image classification model and deployment on Android devices.

### 2.2.1. Class Selection

Only those trees or shrubs over 2 m in height were selected from the TIC database. The Cucarda, Guaylo, and Tilo_or_sauco_blanco classes can be considered as shrubs [42]. A total of fourteen species were finally chosen (Table 1) based on the availability of at least 160 samples, with the highest number of inventoried trees of native (N) or introduced (I) origin.

**Table 1.** Tree species/class selected from the TIC database. (N) Native and (I) introduced species.

| Class Name | Description | Origin | |
|---|---|---|---|
| Alamo blanco<br>*Populus alba* L. | Tree that reaches up to 15 m in height, with straight trunks and greenish white outer bark that cracks over the years [43]. | I | |
| Capuli<br>*Prunus serotina* Ehrh. | Tree that reaches from 8 to 15 m in height and 30 to 50 cm in diameter at breast height (DBH). It has fissured outer bark, alternate branching, and a globose crown [42]. | N | |
| Cepillo<br>*Callistemon lanceolatus* (Sm.) Sweet | Tree from 2 to 7 m in height with dark brown fissured outer bark. It has elongated leaves with a linear shape that are pubescent when young [42]. | I | |
| Cepillo blanco<br>*Melaleuca armillaris* (Sol. ex Gaertn.) Sm. | Tree from 2 to 6 m in height with straight trunk and grayish brown fibrous outer bark. It has very thin leaves of linear shape with an acute apex [44]. | I | |
| Cipres<br>*Cupressus macrocarpa* Hartw. | They can reach 20 m in height with an approximate diameter of about 60 cm. They have a straight trunk and thin bark with longitudinal fissures. Their leaves are very small, scale-like, and aligned in opposite pairs [42]. | I | |

**Table 1.** *Cont.*

| Class Name | Description | Origin | |
|---|---|---|---|
| Cucarda<br>*Hibiscus rosa-sinensis* L. | Shrub with branched stems from 2 to 5 m in height. It has simple leaves that are rounded at the base and elongated towards the apex. Its flowers can be presented in various colors, most commonly red or pink [42]. | I | |
| Eucalipto<br>*Eucaliptus globulus* Labill. | They can reach more than 60 m in height. In some specimens, the outer bark is light brown with a skin-like appearance and it peels off in strips leaving gray or brownish spots on the inner bark [42]. | I | |
| Guabisay<br>*Podocarpus sprucei* Parl | Tree that grows up to 15 m in height in natural environments. It has a straight trunk and grayish brown fissured outer bark. It has simple, alternate, linear leaves with a hard-textured pointed apex [42]. | N | |
| Guaylo<br>*Delostoma integrifolium* D. Don | Tree that reaches up to 6 m in height with a cylindrical trunk and smooth bark. It has simple, opposite elliptic to oblong elliptic leaves with entire margin and terminal inflorescence with a few clustered flowers [42]. | N | |
| Huesito<br>*Pittosporum undulatum* Vent. | Tree from 3 to 5 m in height with grayish brown granular outer bark texture. It has a dense or semi-dense crown of globose or ellipsoidal shape that is light green in color [45]. | I | |
| Ramo de novia<br>*Yucca gigantea* Lem. | Tree from 3 to 6 m in height; when mature, it usually develops several stems. Its trunk has bulges at the base that taper towards the middle part with grayish brown rough outer bark [46]. | I | |
| Sauce<br>*Salix humboldtiana* | Tree that reaches between 5 and 12 m in height and 50 cm in diameter. It has a tortuous trunk with cracked outer bark and a wide irregular crown with alternate branching [42]. | I | |
| Tilo o Sauco blanco<br>*Sambucus mexicana* C. Presl ex DC | Deciduous shrub that reaches up to 3 m or more in height. Its leaves are arranged in opposite pairs [47]. | I | |
| Urapan<br>*Fraxinus excelsior* L. | Tree that grows up to 35 m in height with irregular crown and deciduous foliage. It has opposite, pinnate compound leaves and the leaflets are finely serrated [48]. | I | |

### 2.2.2. Image Selection and Preparation

From the existing species in the TIC, the fourteen with the greatest number of individuals were selected. Of these, Cipres had the least number of images, so the selection process was performed based on the number of images of this class (426), 20% of which (86 images) were reserved for validation. From this set of images, a selection process was performed

through activation maps (Figure 3) in order to use only the images where the relevant pixels (red color in Figure 3) belonged to the actual trees and not to the context (buildings, cars, and others). For each selected image, a corresponding activation map (new image) was generated using the EfficientNetB0 base model from TensorFlow Hub and the Grad-CAM library in Python version 3.9.12. Each generated activation map was visually inspected to determine which images to eliminate based on whether EfficientNetB0 considered the context pixels more relevant than the tree pixels.

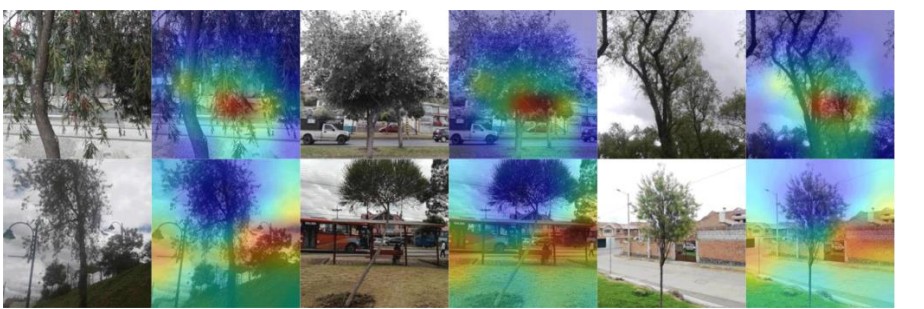

**Figure 3.** Examples of selected and unselected images from their activation maps using Grad-CAM and Inception V3.

Therefore, the number of images for training was reduced, and it was necessary to complete them through synthetic samples obtained by rotation at 90, 180, and 270 degrees (Table 2). This is Data augmentation (DA) which is a technique used to increase the size of the training data set by creating new images from existing ones [30]. Finally, the image size was homogenized through a Python script that performed framing from the center and resized the sample images into $224 \times 224$ pixel images.

**Table 2.** Number of images by class and new rotated images.

| Class | Original Images | Rotation 90° | Rotation 180° | Rotation 270° | Rotated Images | % Images Rotated |
|---|---|---|---|---|---|---|
| Alamo blanco | 295 | 15 | 15 | 15 | 45 | 13% |
| Capuli | 340 | 0 | 0 | 0 | 0 | 0% |
| Cepillo | 340 | 0 | 0 | 0 | 0 | 0% |
| Cepillo blanco | 340 | 0 | 0 | 0 | 0 | 0% |
| Cipres | 271 | 23 | 23 | 23 | 69 | 20% |
| Cucarda | 322 | 18 | 6 | 6 | 6 | 5% |
| Eucalipto | 249 | 31 | 30 | 30 | 91 | 27% |
| Guabisay | 340 | 0 | 0 | 0 | 0 | 0% |
| Guaylo | 289 | 17 | 17 | 17 | 51 | 15% |
| Huesito | 208 | 44 | 44 | 44 | 132 | 39% |
| Ramo de novia | 172 | 56 | 56 | 56 | 168 | 49% |
| Sauce | 340 | 0 | 0 | 0 | 0 | 0% |
| Tilo o sauco blanco | 340 | 0 | 0 | 0 | 0 | 0% |
| Urapan | 340 | 0 | 0 | 0 | 0 | 0% |
| | 4186 | 202 | 187 | 185 | 574 | |

The final photographs of the selected classes were taken between 2017 and 2019 between 07:00 and 18:00, with a higher frequency between 9:00 and 12:00. Approximately 98% of the photographs were acquired with Huawei and Samsung branded smartphones, and the remaining 2% with Apple and Sony phones.

### 2.2.3. Training and Testing Models

Once the training and testing images were defined, the process of model training was performed by TL, for which two base architectures with a standard input size of $224 \times 224$ pixels and 3 bands were selected: RestNet V2 with 101 layers, trained on ImageNet (ILSVRC-2012-CLS) [49], and EfficientNet-Lite, with several versions with different parameters and optimized for edge devices [50]. The two models were implemented under

the TensorFlow Lite model maker library with 20 epochs, which represents the number of times that all training images are fed through the neural network [51]. In the training, we defined the URL of the base model used from TensorFlow Hub and the address of the folder with the validation images. Finally, we fine-tuned the whole model instead of just training the head layer [52].

Two approaches were used to compare the performance of the models with the evaluation dataset: the confusion matrix and the classification reports from the Scikit-learn library (Python). First, the accuracy and Cohen's Kappa metrics allowed us to assess the global reliability of the model. In the second tool, the *Precision*, *Recall*, and *F1-score* (harmonic mean between Precision and Recall) [53] were used to evaluate the performance of each class individually. The equations of these metrics are shown in Equations (1)–(5).

$$Accuracy = \frac{TP + TN}{TP + TN + FP + FN} \tag{1}$$

$$Kappa = \frac{(p_o - p_e)}{(1 - p_e)} \tag{2}$$

$$Precision = \frac{TP}{TP + FP} \tag{3}$$

$$Recall = \frac{TP}{TP + FN} \tag{4}$$

$$F1 - score = \frac{(2 * Precision * Recall)}{Recall + Precision} \tag{5}$$

where *TP* is true positive, *TN* is true negative, *FP* is false positive, and *FN* is false negative in the confusion matrix [53]. For Cohen's Kappa, $p_o$ is the empirical probability of agreement on the label assigned to any sample (the observed agreement ratio) and $p_e$ is the expected agreement when both annotators assign the labels randomly [54].

### 2.2.4. Deployment of the Model in Android Application

The Tensorflow Lite model maker library was chosen for its ease of use in generating models for mobile devices and its integration with the templates proposed in the TensorFlow website [55]. This library saves the trained model in TFLite (TensorFlow lite) format. This format is lightweight, efficient, optimized for mobile devices, and can be used to package trained TensorFlow models for deployment [56].

The mobile devices require an interpreter to execute these models, and uses a combination of hardware acceleration and software optimization to perform the inference based on input data [39,57].

## 3. Results
### 3.1. Learning Curves of the Models

The performance of the proposed models, EfficientNet-Lite and ResNet V2 101, for forest species recognition was tested with Transfer Learning. Figure 4 shows the accuracy and loss learning curves of the EfficientNet-Lite model during training and validation. The training accuracy gradually increased (Figure 4a) and the training and validation loss decreased slowly across all epochs (Figure 4b); in both cases, the accuracy showed fluctuations between epochs 4 and 14 and later stabilized. Figure 5 shows the accuracy and loss of the ResNet-V2-101-based model from epoch three onwards, where the accuracy of the validation data reached a score of 0.9; after which the model did not learn anymore.

Analyzing the accuracy and Cohen's kappa and the Accuracy (Acc) obtained from the validation dataset, the EfficientNet-Lite model performed better than the ResNet V2 101 model (Table 3). We evaluated the model and interpreter statistics in two ways. The first method, called Full Training, involved training the entire model, including layers and

weights, which consumed a significant number of computational resources and time. The second method, called Top Layer Training, only retrained the top layers of the model, and utilized the previously learned weights and features for the other layers.

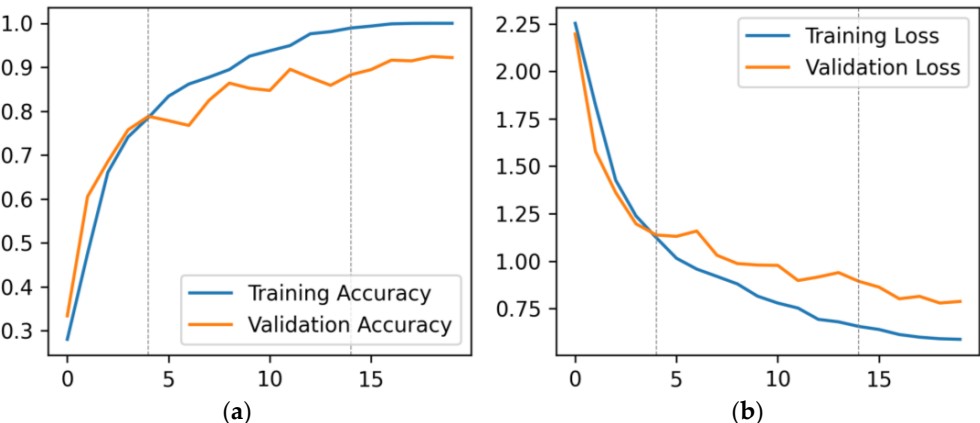

**Figure 4.** The learning curves of the EfficientNet-Lite model using Transfer Learning: (**a**) accuracy learning curves; (**b**) loss learning curves.

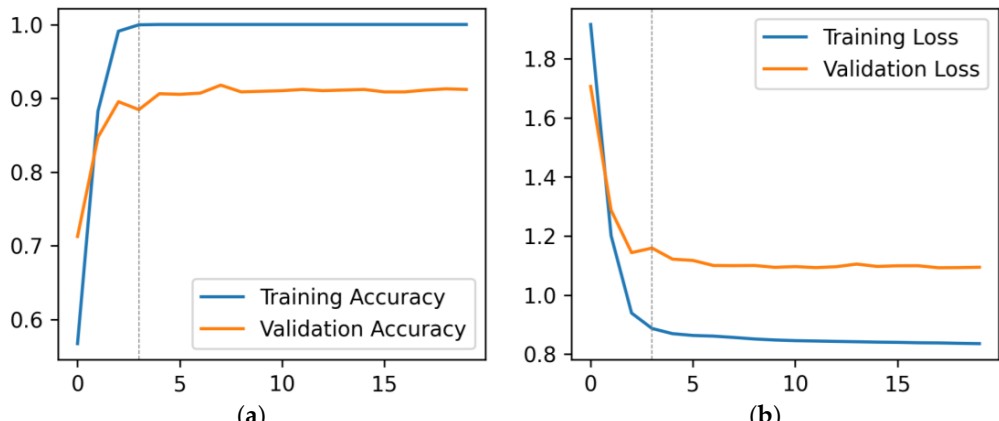

**Figure 5.** Learning curves of the ResNet V2 101 model using Transfer Learning: (**a**) accuracy learning curves; (**b**) loss learning curves.

**Table 3.** Accuracy and kappa of model and interpreter.

| | Full Training | | | | Top Layer Training | | | |
| | Model | | Interpreter | | Model | | Interpreter | |
| **Base Model Name** | **Acc** | **Kappa** | **Acc** | **Kappa** | **Acc** | **Kappa** | **Acc** | **Kappa** |
|---|---|---|---|---|---|---|---|---|
| ResNet V2 101 | 0.912 | 0.905 | 0.801 | 0.785 | 0.750 | 0.731 | 0.678 | 0.653 |
| EfficientNet-Lite | 0.922 | 0.916 | 0.770 | 0.752 | 0.780 | 0.763 | 0.648 | 0.621 |

We assessed the predictions made by the TensorFlow Lite Interpreter, and we found that ResNet V2 101 was the best choice, even though its reliability decreased compared to the original model.

### 3.2. Class Accuracy with the Best Model

Attending to the F1-score of the model based on ResNet V2 101, there were four highly reliable classes: Ramo de novia, Cipres, Cepillo blanco, and Sauce (Figure 6). The loss of F1-score between the model and the interpreter was not homogeneous across all classes. Note that it was higher for the Cepillo and Urapan classes compared to the others.

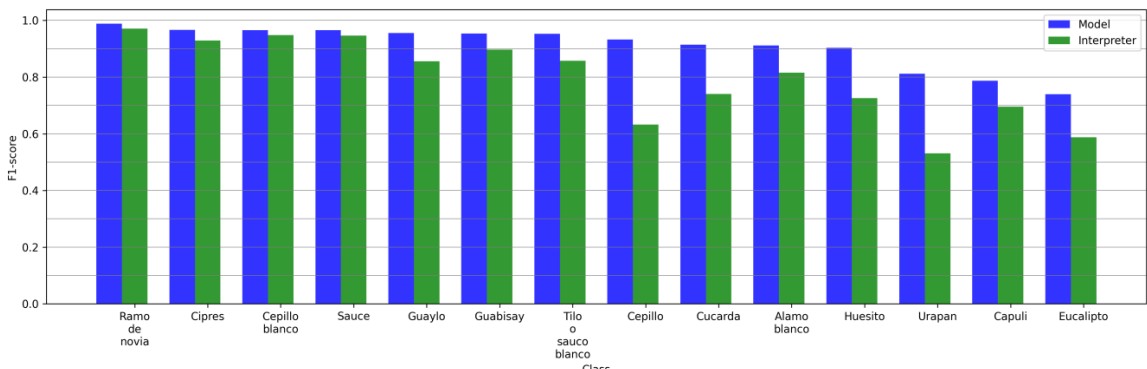

**Figure 6.** F1-scores for all classes considered using ResNet V2 101 model base.

Using the full report of the classification model (Table 4) and the interpreter (Table 5), a total of eight classes exceeded an F1-score of 0.8. The classes Eucalipto, Capuli, and Urapan were the most difficult to classify. The confusion matrices (model and interpreter) with the greatest confusion occurred between the Eucalipto and Capuli classes. In the case of the interpreter, in addition to the previous one, other cases also occurred, such as the confusion of Urapan with Eucalipto, and Cepillo with Cucarda.

**Table 4.** Confusion matrix and classification report using ResNet V2 101 base model.

| | | | | | | | Prediction | | | | | | | | | | | |
|---|---|---|---|---|---|---|---|---|---|---|---|---|---|---|---|---|---|---|
| | | 1 | 2 | 3 | 4 | 5 | 6 | 7 | 8 | 9 | 10 | 11 | 12 | 13 | 14 | Precision | Recall | F1-Score |
| | 1 | 82 | 0 | 0 | 0 | 1 | 0 | 0 | 0 | 1 | 0 | 0 | 1 | 0 | 1 | 0.87 | 0.95 | 0.91 |
| | 2 | 3 | 74 | 0 | 0 | 0 | 0 | 1 | 1 | 1 | 3 | 0 | 0 | 1 | 2 | 0.73 | 0.86 | 0.79 |
| | 3 | 0 | 2 | 76 | 2 | 1 | 4 | 0 | 1 | 0 | 0 | 0 | 0 | 0 | 0 | 0.99 | 0.88 | 0.93 |
| | 4 | 0 | 0 | 0 | 84 | 0 | 0 | 0 | 0 | 1 | 0 | 0 | 1 | 0 | 0 | 0.95 | 0.98 | 0.97 |
| | 5 | 0 | 0 | 0 | 0 | 86 | 0 | 0 | 0 | 0 | 0 | 0 | 0 | 0 | 0 | 0.93 | 1.00 | 0.97 |
| | 6 | 0 | 1 | 0 | 0 | 0 | 80 | 0 | 0 | 3 | 0 | 0 | 0 | 0 | 2 | 0.90 | 0.93 | 0.91 |
| **Real** | 7 | 8 | 19 | 0 | 0 | 0 | 0 | 54 | 0 | 0 | 0 | 1 | 0 | 0 | 4 | 0.90 | 0.63 | 0.74 |
| | 8 | 0 | 0 | 0 | 0 | 0 | 0 | 1 | 83 | 0 | 1 | 0 | 1 | 0 | 0 | 0.94 | 0.97 | 0.95 |
| | 9 | 0 | 0 | 0 | 0 | 0 | 0 | 0 | 0 | 86 | 0 | 0 | 0 | 0 | 0 | 0.91 | 1.00 | 0.96 |
| | 10 | 0 | 1 | 1 | 0 | 0 | 0 | 0 | 1 | 2 | 75 | 0 | 0 | 1 | 5 | 0.94 | 0.87 | 0.90 |
| | 11 | 0 | 0 | 0 | 1 | 0 | 0 | 0 | 0 | 0 | 0 | 85 | 0 | 0 | 0 | 0.99 | 0.99 | 0.99 |
| | 12 | 0 | 1 | 0 | 0 | 0 | 0 | 1 | 0 | 0 | 0 | 0 | 84 | 0 | 0 | 0.95 | 0.98 | 0.97 |
| | 13 | 0 | 3 | 0 | 0 | 0 | 1 | 0 | 1 | 0 | 0 | 0 | 0 | 80 | 1 | 0.98 | 0.93 | 0.95 |
| | 14 | 1 | 1 | 0 | 1 | 1 | 4 | 4 | 3 | 0 | 1 | 1 | 0 | 1 | 69 | 0.82 | 0.80 | 0.81 |

(1) Alamo Blanco, (2) Capuli, (3) Cepillo, (4) Cepillo blanco, (5) Cipres, (6) Cucarda, (7) Eucalipto, (8) Guabisay, (9) Guaylo, (10) Huesito, (11) Ramo de novia, (12) Sauce, (13) Tilo o sauco blanco, (14) Urapan.

**Table 5.** Confusion matrix and classification report using ResNet V2 101 interpreter.

| | | | | | | | Prediction | | | | | | | | | | | |
|---|---|---|---|---|---|---|---|---|---|---|---|---|---|---|---|---|---|---|
| | | 1 | 2 | 3 | 4 | 5 | 6 | 7 | 8 | 9 | 10 | 11 | 12 | 13 | 14 | Precision | Recall | F1-Score |
| | 1 | 84 | 0 | 0 | 0 | 1 | 0 | 0 | 0 | 0 | 0 | 0 | 0 | 0 | 1 | 0.70 | 0.98 | 0.82 |
| | 2 | 6 | 64 | 0 | 0 | 0 | 1 | 11 | 1 | 3 | 0 | 0 | 0 | 0 | 0 | 0.65 | 0.74 | 0.70 |
| | 3 | 4 | 5 | 43 | 4 | 2 | 13 | 4 | 4 | 2 | 0 | 2 | 0 | 0 | 3 | 0.86 | 0.50 | 0.63 |
| | 4 | 0 | 0 | 0 | 83 | 0 | 0 | 0 | 1 | 0 | 0 | 0 | 1 | 1 | 0 | 0.93 | 0.97 | 0.95 |
| | 5 | 1 | 0 | 0 | 0 | 85 | 0 | 0 | 0 | 0 | 0 | 0 | 0 | 0 | 0 | 0.88 | 0.99 | 0.93 |
| | 6 | 2 | 1 | 0 | 0 | 0 | 77 | 1 | 0 | 5 | 0 | 0 | 0 | 0 | 0 | 0.63 | 0.90 | 0.74 |
| **Real** | 7 | 8 | 17 | 1 | 0 | 2 | 1 | 52 | 0 | 0 | 0 | 1 | 0 | 1 | 3 | 0.57 | 0.60 | 0.59 |
| | 8 | 2 | 0 | 0 | 0 | 0 | 1 | 1 | 78 | 0 | 0 | 0 | 1 | 0 | 3 | 0.89 | 0.91 | 0.90 |
| | 9 | 5 | 0 | 0 | 0 | 0 | 1 | 0 | 0 | 80 | 0 | 0 | 0 | 0 | 0 | 0.79 | 0.93 | 0.86 |
| | 10 | 2 | 2 | 0 | 0 | 1 | 11 | 3 | 1 | 9 | 49 | 1 | 0 | 0 | 7 | 1.00 | 0.57 | 0.73 |
| | 11 | 0 | 0 | 0 | 1 | 0 | 0 | 0 | 0 | 0 | 0 | 85 | 0 | 0 | 0 | 0.96 | 0.99 | 0.97 |
| | 12 | 0 | 1 | 0 | 0 | 0 | 0 | 0 | 2 | 0 | 0 | 0 | 79 | 0 | 4 | 0.98 | 0.92 | 0.95 |
| | 13 | 2 | 7 | 1 | 0 | 0 | 7 | 1 | 1 | 0 | 0 | 0 | 0 | 66 | 1 | 0.97 | 0.77 | 0.86 |
| | 14 | 4 | 1 | 5 | 1 | 6 | 10 | 18 | 0 | 2 | 0 | 0 | 0 | 0 | 39 | 0.64 | 0.45 | 0.53 |

(1) Alamo Blanco, (2) Capuli, (3) Cepillo, (4) Cepillo blanco, (5) Cipres, (6) Cucarda, (7) Eucalipto, (8) Guabisay, (9) Guaylo, (10) Huesito, (11) Ramo de novia, (12) Sauce, (13) Tilo o sauco blanco, (14) Urapan.

Evaluating the classification probabilities assigned to each of the 86 images per class (Figure 7), the classes with the best F1-score ranking also presented a high probability of correctness. The least favorable probabilities were for Eucalipto, which tended to be confused with the Capuli class. For the Urapan class, the probability of success was lower.

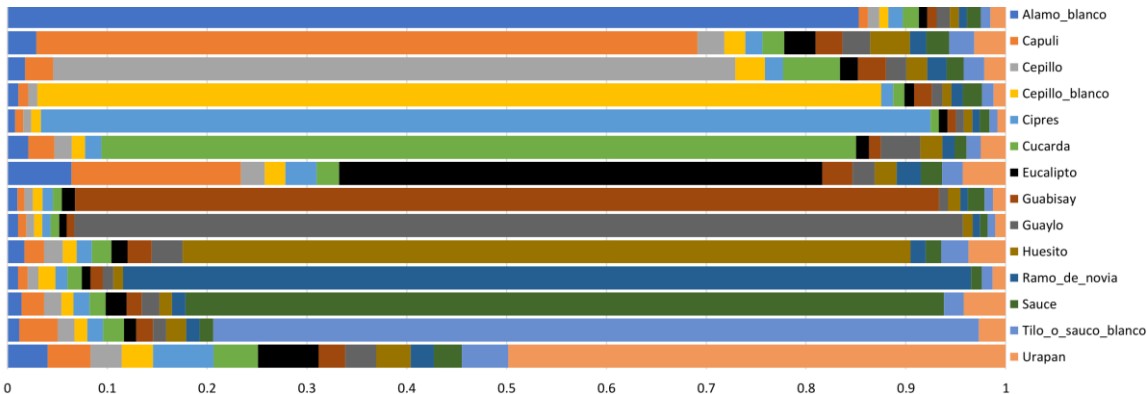

**Figure 7.** Sum of probabilities by classes of the 86 testing images per class.

### 3.3. Model Deployment in Android Devices

The mobile application installer and model in mobile format (TFLite) can be downloaded from the website https://gis.uazuay.edu.ec/ (accessed on 15 March 2023). When performing the real-time prediction, the tree characteristics vary due to wind and light effects; the four highest-ranking possibilities are presented in the screen. When similar values exist, the tree cannot be classified correctly at that moment, and a change of the camera position is needed. When there is a significant difference between the first and second predictions, then there is more certainty in the result. Additionally, the proximity of the camera sensor to the object influences the prediction, as in the example shown in Figure 8, where the probability of the prediction varied from 51.2% to 89.5%.

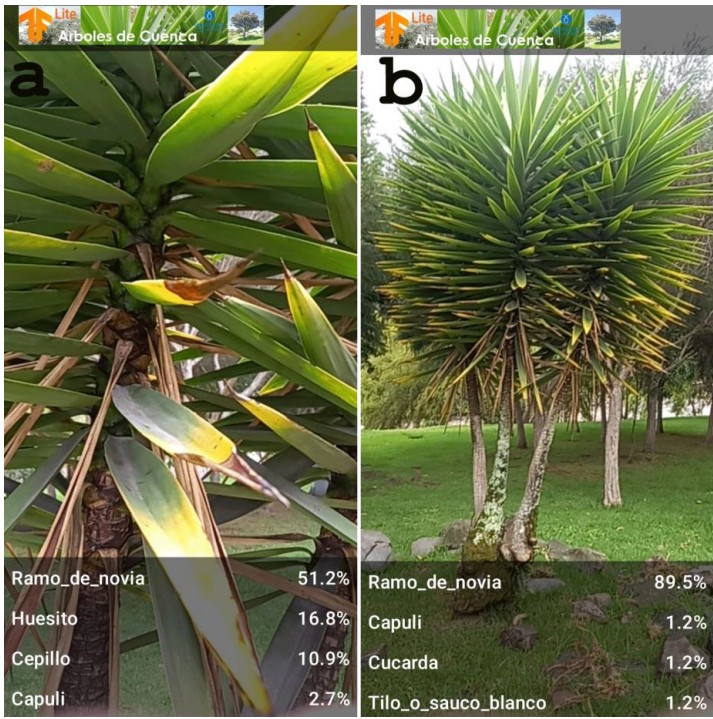

**Figure 8.** Real-time example classification of the Ramo de Novia class. (**a**) At a distance of approximately one meter, and (**b**) at a distance of approximately five meters.

## 4. Discussion

In this work, two different types of results were obtained, from a model and interpreter (using smartphones). In both, the best accuracy and kappa metrics were obtained using ResNet V2 101 as the base model (Transfer Learning approach), with values of 0.912 and 0.905 in the model and 0.801 and 0.785 in the interpreter, respectively. This accuracy is similar to those obtained in works such as [39,53,58]. The TensorFlow Lite format, used in the interpreter, decreased the number of operations and affected the accuracy of the interpreter model.

The main difference between this work and others is that the images of TIC were taken entirely with smartphones, generally of the whole tree or shrub, and under different circumstances that add a high variability such as shadows, amount of light, distance to the object, and time of day, among others. For example, in [39], the images of leaves were taken using a Sony DSC—Rx100/13 and used for the identification of plant diseases. These images were at a resolution of 20.2 megapixels. On the other hand, the images in [58] focused on specific characteristics such as leaves and bark from different species.

In Transfer Learning [28,39], it was concluded that our dataset's best-performing model for Transfer Learning was ResNet V2 101 that was fully retrained [59] (not only the classification layer on top). ResNet is an architecture that was previously used in similar applications such as [39] with 50 layers, while the one used in this work was 101 layers deep. From existing ResNet architectures, ResNet V2 101 had high reliability in other works with small datasets for the classification of plant leaf diseases [22] or in works using drone images, such as [59–61]. However, the ResNet V2 101 model has many more parameters than simpler models [62], which significantly increases the memory and processing requirements needed to train the model. In this work, controlling the number of images used for training was necessary due to computational constraints such as limited memory, CPU usage, and non-GPU availability, as well as processor capacity. In order to limit the number of images used, we chose to use only the image rotation data augmentation technique, used in other works such as [33,39].

Using the F1-score as a reference, from the fourteen classes evaluated in the model, ten exceeded a value of 0.9 and four in the interpreter. In the model, the classes with the highest value of F1-score were Guaylo, Cepillo blanco, Cipres, Sauce, and Ramo de Novia. For the interpreter, the best classes were Cipres, Cepillo blanco, Sauce, and Ramo de Novia.

The morphological characteristics of the Ramo de Novia class differentiated it from the other evaluated species, making this the most reliable class. The classes with the lowest reliability were Eucalipto and Capuli in the model, and Urapan and Eucalipto in the interpreter. As a test, the Capuli class was re-evaluated with another set of training and validation photographs to determine if this factor affected its reliability, obtaining similar results to the original classification. The morphological characteristics of this tree make it susceptible to confusion with other species. For this class, we propose to specifically evaluate the leaves [63], which are the organ most used in the identification of plants, or fruits for their identification. Other options to explore are the use of images of flowers [7] or bark [19].

These highest and lowest accuracy classes will be analyzed in future work using feature extraction such as color, texture, shape [64], which are commonly used for leaf feature extraction, to describe the characteristics that make them identifiable to a CNN. In addition, other data augmentation techniques may be included such as Laplacian sharpening, Gaussian blur augmentation, contrast enhancement, shifting, cropping, and zooming [65,66].

Even though the model's reliability was greater than that of the interpreter, we believe that further work should be able to improve the interpreter model, for example, using TensorFlow Model Optimization Toolkit [67]. This would be desirable in those circumstances where tree identification must be carried out in real time and the mobile device does not have Internet connection.

## 5. Conclusions

This paper presented an evaluation of two base models, ResNet V2 and EfficientNet-Lite, in a TensorFlow Lite model maker library for the recognition of fourteen classes of trees using the images of the TIC dataset. The performance of model with ResNet V2 101 was superior to that obtained with EfficientNet-Lite and the species Cepillo blanco, Cipres, Sauce, Guabisay, and Ramo de novia presented the highest values of accuracy, Kappa, and F1-score in the classification. The final model had an accuracy of 0.912 and Kappa of 0.905, and with the TensorFlow Lite interpreter having an accuracy of 0.801 and Kappa of 0.785.

In the future, new tree species will be tested so they can be added to the application, generating a new classification model. It is expected that the mobile application will help non-expert users in the identification of tree species through smartphones, and it is considered a starting point for the creation of a support for tree's inventories generation and maintenance in urban environments.

**Author Contributions:** Conceptualization, D.P.-P. and L.Á.R.; methodology, D.P.-P.; investigation, D.P.-P. and E.B.-L.; writing—original draft preparation, D.P.-P. and E.B.-L.; supervision, L.Á.R. All authors have read and agreed to the published version of the manuscript. Authorship is limited to those who have contributed substantially to the work reported.

**Funding:** This research was funded by the University of Azuay in the context of investigation project 2020-0125 denominated "Caracterización de unidades forestales a partir de datos espectrales, espaciales y de relieve a distintas escalas. Aplicación a los bosques andinos del cantón Cuenca (Ecuador), Fase 2".

**Data Availability Statement:** These data are a private dataset and can be solicited from the Instituto de Estudios de Regimen Seccional del Ecuador (IERSE) at http://ierse.uazuay.edu.ec (accessed on 30 March 2022).

**Conflicts of Interest:** The authors declare no conflict of interest.

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
