# Peer review of "Tree Species Identification in Urban Environments Using TensorFlow Lite and a Transfer Learning Approach"

_forests, doi:10.3390/f14051050_

Round 1
Reviewer 1 Report
The article is very well-written and easy to read. The literature review performed in the introduction section is adequate for the purposes of the article. The methodology is appropriate and clearly described. Congratulations on the work developed.
I have two suggestions and one question.
P1L32 I would not use the word “forest”.
A reference to including shrubs in the study should be added in the methodology section (e.g. 2.2.1).
How can the results be improved for some species classes, such as Eucalyptus and Capuli, that were not as well distinguished? Shouldn't other tree characteristics, such as leaves, be considered in addition to the tree structure? Perhaps the authors could elaborate on this aspect in their discussion.
Author Response
Response to Reviewer 1 Comments
Dear Reviewer
On behalf of the entire research team, I thank you for your valuable suggestions and observations, which helped to improve the quality of our manuscript.
Below, we respond to each of your comments:
Point 1: P1L32 I would not use the word “forest”.
Response 1: Thank you very much for your suggestion. We have modified the original sentence (P1L32):
Urban Tree Inventories (UTI) provide information about species, location and photography of each tree [2,3], which will allow a sustainable forest management and monitoring of urban trees and gardens.
Point 2: A reference to including shrubs in the study should be added in the methodology section (e.g. 2.2.1)
Response 2:
The reference was added in section 2.2.1 as follows (P4 L143-144):
Only those trees or shrubs over 2 m height were selected from the TIC database. The Cucarda, Guaylo and Tilo_o_sauco_blanco classes can be presented as shrubs [42].
[42] Minga, D.; Verdugo, A. Árboles y Arbustos de Los Ríos de Cuenca; Cuenca, 2016; ISBN 978-9978-325-42-1.
Point 3 (Question): How can the results be improved for some species classes, such as Eucalyptus and Capuli, that were not as well distinguished? Shouldn't other tree characteristics, such as leaves, be considered in addition to the tree structure? Perhaps the authors could elaborate on this aspect in their discussion.
To improve the results for the lower reliability classes, we are considering future work consisting of the following:
- We will evaluate if there are new photographs of these species Capuli and Eucalyptus in the Tree Inventory of Cuenca (TIC) that we can use to replace the current ones.
- We plan to evaluate and compare other Data Augmentation techniques to increase the number of images for training and validation. Additionally, we can evaluate other TensorFlow Hub models that support 224x224 sized photographs.
- For species with lower reliability, we can generate a parallel model that explores only photographs of flowers, fruits, bark, which is already done by most tree identification applications. If we obtain favorable results with these photographs, we propose to modify the TIC photo collection process to add a specific photograph of this elements.
We have added these aspects in Discussion section (P11 L287 – P12 L336).
Once again, we are grateful for the revision of our paper, which has improved its quality.
Best regards.
Diego Pacheco Prado (corresponding author)
Universidad del Azuay (Ecuador) - Universidad Politécnica de Valencia (Spain)

Reviewer 2 Report
Deep learning has been applied plant identification successfully. In this paper, ResNet V2101 and EfficientNet-Lite were used for recognition of fourteen tree species in urban environments. The experimental results showed that ResNet V2101 performed best with an accuracy of 0.912, which validated the usefulness of deep learning methods. However, it is necessary to explain why only rotation was selected for data augmentation and why the number of images of each class after data augmentation was the same.
Author Response
Response to Reviewer 2 Comments
Dear Reviewer
On behalf of the entire research team, I thank you for your valuable suggestions and observations, which helped to improve the quality of our manuscript.
Below, we respond to each of your comments:
Point 1: Deep learning has been applied plant identification successfully. In this paper, ResNet V2101 and EfficientNet-Lite were used for recognition of fourteen tree species in urban environments. The experimental results showed that ResNet V2101 performed best with an accuracy of 0.912, which validated the usefulness of deep learning methods. However, it is necessary to explain why only rotation was selected for data augmentation and why the number of images of each class after data augmentation was the same.
Response 1: Thank you very much for your suggestion. We added a paragraph into Discussion section supported by bibliographical references (P11L300 – P12L312).
In Transfer Learning, ResNet is an architecture previously used in similar applications such as [39], although with 50 layers deep, while the one used in this work is 101 layers deep. ResNet V2 101 had high reliability in other work with small datasets for the classification of plant leaf diseases [22], or in works using drone images such as [62–64]. However, the ResNet V2 101 model selected has many more parameters than simpler models [65], which significantly increases the memory and processing requirements needed to train the model. Controlling the number of images used for training was necessary in this work due to computational constraints such as limited memory, CPU usage, and non-GPU availability, as well as processor capacity. In order to limit the number of images used, we chose to use only the image rotation data augmentation technique, used in other works as [33,39].
[33] Figueroa-Mata, G.; Mata-Montero, E. Using a Convolutional Siamese Network for Image-Based Plant Species Identification with Small Datasets. Biomimetics 2020, 5, doi:10.3390/biomimetics5010008.
[39] Reda, M.; Suwwan, R.; Alkafri, S.; Rashed, Y.; Shanableh, T. AgroAId: A Mobile App System for Visual Classification of Plant Species and Diseases Using Deep Learning and TensorFlow Lite. Informatics 2022, 9, doi:10.3390/informatics9030055
[65] Chen, L.; Li, S.; Bai, Q.; Yang, J.; Jiang, S.; Miao, Y. Review of Image Classification Algorithms Based on Convolutional Neural Networks. Remote Sens. 2021, 13, 1–51, doi:10.3390/rs13224712.
Once again, we are grateful for the revision of our paper, which has improved its quality.
Best regards.
Diego Pacheco Prado (corresponding author)
Universidad del Azuay (Ecuador) - Universidad Politécnica de Valencia (Spain)

Reviewer 3 Report
This paper used ResNet V2 and EfficientNet-Lite modes to recognize fourteen classes of trees using the images of TIC dataset. They found the performance of ResNet V2 101 model was superior to EfficientNet-Lite. The paper also identified the species that have higher classification accuracy. Although the paper showed clear objectives, I cannot capture the value of a research paper should behave. The value of science and necessary innovation has not been reflected. I think the paper might be not suite to be published in Forest according to its current version, though I did not provide any specific comments. I think the paper should pay more attention to why some species can be better recognized and why some not. What are the critical characteristics that the two methods extract? How the images impacted the accuracies in different models? The discussion section should be more focused on the main results, the current text is somewhat dispersed.
Author Response
Response to Reviewer 3 Comments
Dear Reviewer
On behalf of the entire research team, I thank you for your valuable suggestions and observations, which helped to improve the quality of our study.
Below, We respond to each of your comments:
Point 1: This paper used ResNet V2 and EfficientNet-Lite modes to recognize fourteen classes of trees using the images of TIC dataset. They found the performance of ResNet V2 101 model was superior to EfficientNet-Lite. The paper also identified the species that have higher classification accuracy. Although the paper showed clear objectives, I cannot capture the value of a research paper should behave. The value of science and necessary innovation has not been reflected. I think the paper might be not suite to be published in Forest according to its current version, though I did not provide any specific comments.
Response 1: We respect your criteria, however we consider that our work is suitable for publication in Forest, since it develops an exploratory process to determine if the photographs of the Tree Inventory of Cuenca (TIC) could be used to train a tree identification model. These photographs were taken entirely with smartphones, and under different circumstances that add a high variability such as: shadows, amount of light, distance to the object, time of day, among others. Although it is not completely focused on forestry, as a result a specific application is developed focused on the detection of plant species with a computational approach, which opens the way for the development of new research in the study area, which has no studies of this type.
In addition, as there are several models of base architectures for Transfer Learning, it seems appropriate to first determine one that best fits our data, that maximizes its efficiency with small images (224x224 pixels) but still maintains a high degree of reliability. With this we can project ourselves to further improve the model, without requiring (relatively) high computing power.
A key aspect for us is that this knowledge is transferred to the non-expert user, through a mobile application, which we have already evaluated and redesigned to be improved, in order to expand its functionality in future work.
Once we have defined a workflow that we know it works, we can explore the points addressed in your comment, such as image quality, species with better reliability and critical characteristics, as well as expand to other existing species in the TIC in further research.
Point 2. I think the paper should pay more attention to why some species can be better recognized and why some not. What are the critical characteristics that the two methods extract.
Response 2: Thanks for your comment. Indeed, we can now focus our efforts on determining what factors influenced some classes to identify themselves better than others. We have added this point to the discussion (P12L317-P12L325)
Point 3: How the images impacted the accuracies in different models. The discussion section should be more focused on the main results, the current text is somewhat dispersed.
Response 3: With all the points raised, the discussion was redefined (P11L287 – P12L336)
Once again, we are grateful for the revision of our paper, which has improved its quality.
Best regards.
Diego Pacheco Prado (corresponding author)
Universidad del Azuay (Ecuador) - Universidad Politécnica de Valencia (Spain)
